# Polarity protein SCRIB interacts with SLC3A2 to regulate proliferation and tamoxifen resistance in ER+ breast cancer

Yasuhiro Saito [1✉], Shiori Matsuda[1], Naomi Ohnishi[2], Keiko Endo[1], Sanae Ashitani[1], Maki Ohishi[1], Ayano Ueno[1], Masaru Tomita[1], Koji Ueda [2], Tomoyoshi Soga [1✉] & Senthil K. Muthuswamy [3✉]

Estrogen receptor (ER) positive breast cancer represents 75% of all breast cancers in women. Although patients with ER+ cancers receive endocrine therapies, more than 30% develop resistance and succumb to the disease, highlighting the need to understand endocrine resistance. Here we show an unexpected role for the cell polarity protein SCRIB as a tumor-promoter and a regulator of endocrine resistance in ER-positive breast cancer cells. SCRIB expression is induced by estrogen signaling in a MYC-dependent manner. SCRIB interacts with SLC3A2, a heteromeric component of leucine amino acid transporter SLC7A5. SLC3A2 binds to the N-terminus of SCRIB to facilitate the formation of SCRIB/SLC3A2/LLGL2/SLC7A5 quaternary complex required for membrane localization of the amino acid transporter complex. Both SCRIB and SLC3A2 are required for cell proliferation and tamoxifen resistance in ER+ cells identifying a new role for the SCRIB/SLC3A2 complex in ER+ breast cancer.

[1] Institute for Advanced Biosciences, Keio University, 246-2 Mizukami, Kakuganji, Tsuruoka, Yamagata 997-0052, Japan. [2] Cancer Proteomics Group, Cancer Precision Medicine Center, Japanese Foundation for Cancer Research, 3-8-31, Ariake, Koto, Tokyo, Japan. [3] Department of Medicine and Pathology, Cancer Research Institute, Beth Israel Deaconess Medical Center, Harvard Medical School, 3 Blackfan Circle, Boston, MA 02215, United States. ✉email: ysaito@ttck.keio.ac.jp; soga@sfc.keio.ac.jp; smuthusw@bidmc.harvard.edu

Breast cancer is a significant cause of death globally, and approximately 75%[1] of breast cancers are driven by aberrant expression of estrogen receptor (ER)[2]. Although patients with ER-positive (ER+) disease are eligible for endocrine therapy, the development of resistance and metastatic progression is a leading cause of death for breast cancer patients. Identifying new vulnerabilities for combating resistance to anti-estrogen therapy is an essential topic in breast cancer research.

Metabolic reprogramming enables cancer cells to continue growing under stressful environments, including a lack of nutrients[3] and therapeutic drug treatments[4]. Deregulated amino acid uptake[3] and aberrant cell surface expression of amino acid transporters[4] are known to occur in cancer cells, and its role in ER+ breast cancer is beginning to be understood. However, the molecular mechanisms that regulate the increase in cell surface levels of amino acid transporters are poorly understood.

Cell polarity proteins Scribble (scrib) and Lethal giant larvae (lgl) were identified as tumor suppressor genes in Drosophila because the loss of function mutations lead to uncontrolled proliferation of epithelial cells[5]. Loss of function mutation of mammalian homolog genes LLGL2 (lgl) and SCRIB (scrib) does not result in neoplastic growth, suggesting divergence in protein function in mammalian epithelial cells[6]. Hence a direct relationship between tumor suppression and SCRIB or LLGL2 proteins is controversial in mammals.

We and others have reported tumor-promoting roles of LLGL2 and SCRIB in breast cancer cells[7–9]. SCRIB Pro 305 Leu mutant (SCRIB P305L), which fails to locate plasma membrane, promotes cell proliferation in mammary epithelial cells in vivo[9], and downregulation of SCRIB attenuates the tumor-growth ability in ER-negative (ER−) breast cancer cells[9]. We recently reported that LLGL2 overexpression promotes cell proliferation under nutrient stress conditions and regulates tamoxifen resistance in ER+ breast cancer cells[7]. Although SCRIB and LLGLs (LLGL1 and LLGL2) are scaffolding proteins that interact to regulate apical-basal polarization in mammalian epithelium[10], whether the SCRIB and LLGL polarity module regulates the biology of ER+ breast cancer is unknown.

Here we report that SCRIB promotes cell proliferation in ER+ breast cancer cells in culture and in vivo. SCRIB interacts with SLC3A2, a heteromeric component of the L-type amino acid transporter 1 (LAT1), SLC7A5, and regulates cell surface transport. Unexpectedly, SCRIB and SLC3A2 form a quaternary complex with LLGL2-SLC7A5 to promote membrane assembly of the SLC7A5/SLC3A2 amino acid transporter complex, which is needed for leucine uptake and proliferation of ER+ breast cancer cells in culture and in vivo. SCRIB expression was stimulated by estrogen-induced MYC-MAX binding to promoter/enhancer in the SCRIB gene in the downstream of ER signaling. SLC3A2, SCRIB, and MYC were upregulated during tamoxifen resistance and required for maintenance of the resistance phenotype.

## Results

### SCRIB promotes cell proliferation in ER+ breast cancer cells.
We found that the SCRIB gene is amplified in 14.42% of breast cancer patients in the TCGA dataset, and the gene amplification correlates with SCRIB mRNA expression in the TCGA breast cancer dataset (Supplementary Fig. 1a). High levels of SCRIB mRNA expression correlated with poor clinical survival in the patients with ER+/progesterone receptor-positive (PR+) status (Fig. 1a) whereas SCRIB mRNA expression did not correlate with poor clinical survival in the patients with ER−/PR+, ER+/PR−, ER−/PR−, HER2+, and ER−/PR−/HER2− status (Supplementary Fig. 1 b–f), leading us to investigate the pathological roles of SCRIB in ER+ breast cancer cells.

Unlike LLGL2[7], SCRIB protein was not differentially expressed and was detected in all types of breast tumors in ER+ and ER− breast cancer patients' tissues (Supplementary Fig. 1g) and cell lysates (Supplementary Fig. 1h). Interestingly, as observed for LLGL2, short hairpin RNA (shRNA)-mediated knockdown of SCRIB in both MCF-7 and T47D cells inhibited cell proliferation under nutrient-stress conditions of serum-free DMEM/F12 culture medium supplemented with B27 and 20 ng/ml epidermal growth factor (EGF) in suspension and 2D culture condition (Fig. 1b–e and Supplementary Fig. 1i, j). Knockdown of SCRIB did not affect cell viability in MCF-7 and T47D cells (Supplementary Fig. 1k–n). The potential off-target effect of shRNA was ruled out using an independent SCRIB shRNA (Supplementary Fig. 1o–r). In addition to the cell proliferation defects observed in culture, orthotopic transplantation of MCF-7 cells demonstrated significant inhibition of in vivo tumor growth ability in cells lacking SCRIB (Fig. 1f and Supplementary Fig. 1s). In addition, overexpression of HA-tagged wild-type SCRIB in ER+ breast cancer cells promoted cell proliferation in suspension culture conditions (Supplementary Fig. 1t–w). Unlike LLGL2, SCRIB-KD suppressed cell proliferation in the culture medium supplemented with 10% FBS (Supplementary Fig. 1x). These results led us to conclude that SCRIB promotes cell proliferation in ER+ breast cancer cells and to examine the pathological roles of SCRIB under nutrient stress conditions.

### SCRIB controls leucine uptake in ER+ breast cancer cells.
Driven by our previous observation on LLGL2 to regulate cellular metabolome in ER+ breast cancer cells, we analyzed SCRIB-KD-induced changes in cellular metabolites in ER+ breast cancer cells. Metabolome analysis by capillary-electrophoresis time of flight mass spectrometry (CE-TOFMS)[11] revealed a substantial decrease in 69 metabolites in MCF-7 cells and 126 in T47D cells (Fig. 1g, Supplementary Fig. 2a–c). Fifty-seven metabolites were downregulated in both MCF-7 and T47D cells (Fig. 1g). Furthermore, among the fifty-seven metabolites downregulated in SCRIB-KD cultured cells, twenty-five metabolites, including leucine, were downregulated by SCRIB-KD in vivo tumors (Fig. 1h and Supplementary Fig. 2d–f). We had demonstrated previously that the growth of ER+ breast cancer cells highly relies on leucine uptake under our nutrient stress culture condition[7]. Thus, we investigated the role of leucine in supporting the growth of SCRIB-KD cells. Interestingly, the presence of 10x leucine/glutamine in the culture media rescued the proliferation defect of SCRIB-KD cells with no detectable effect on the growth of control cells (Fig. 1i). These results demonstrate that, like LLGL2, SCRIB controls the proliferation of ER+ breast cancer by regulating leucine uptake.

### SCRIB and SLC3A2 are part of a quaternary complex with LLGL2 and SLC7A5.
SCRIB is a scaffold protein and localizes at both plasma membrane and cytosol[10]. To gain insight into how SCRIB promotes proliferation in ER+ breast cancer cells, we performed proximity-dependent biotin identification (BioID)[12] for an unbiased interactome analysis to find the proteins that interact with SCRIB. We prepared the SCRIB gene fused with biotin ligase (BirA*) at the N-terminal region of SCRIB and expressed it in MCF-7 cells (Supplementary Fig. 3a). Like wild-type SCRIB, overexpression of BirA*-SCRIB also promoted cell proliferation, indicating that the functional properties of SCRIB were not impacted by the BirA* fusion (Supplementary Fig. 3b). The BioID analysis identified 26 previously reported SCRIB-binding proteins (Supplementary Fig. 3c)[10]. In addition, we observed SLC7A5 and SLC3A2 as novel SCRIB interacting proteins[13,14]. Endogenous SCRIB-SLC3A2 interaction was readily detectable by immunoprecipitation (Fig. 2a and Supplementary

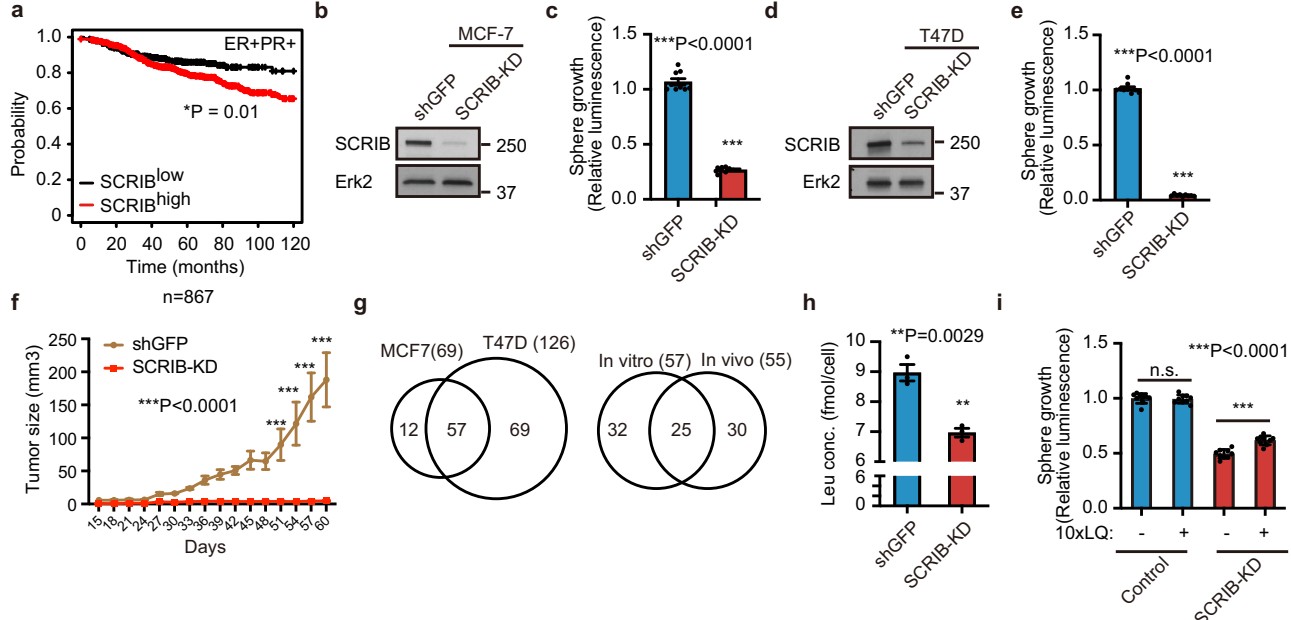

**Fig. 1 SCRIB promotes cell proliferation by regulating leucine uptake in ER+ breast cancer. a** Kaplan–Meier plot of ER + PR+ breast cancer patients' survival. **b** Knockdown of SCRIB in MCF-7 cells. **c** Cell viability assay of SCRIB-KD cells. **d** Knockdown of SCRIB in T47D cells. **e** Cell viability assay of SCRIB-KD T47D cells. **f** Tumor growth of SCRIB-KD MCF-7 cells in NSG mice. **g** Venn diagram representation of the number of intracellular metabolites downregulated by SCRIB-KD between MCF-7 and T47D cells (left) and between SCRIB-KD cultured cells (in vitro) and SCRIB-KD tumors (in vivo) (right). **h** Relative concentration of intracellular leucine in MCF-7 cells. **i** Sphere growth of SCRIB-KD MCF-7 cells in 10X leucine/glutamine (LQ) medium. Data **c**, **e**, **h**, and **i** are shown as mean ± s.e.m.; **c**, **e**, and **i**; $n = 9$, **h**; $n = 3$, **f**; $n = 5$. Statistical analysis was conducted by $t$-test (**c**, **e**, **h**), one-way ANOVA followed by Tukey's posttest (**i**), and two-way ANOVA followed by Sidak's posttest (**f**).

Fig. 3d). Also, SCRIB co-localized with SLC3A2 at the plasma membrane (Fig. 2b and Supplementary Fig. 3e), providing further support to the BioID results. SLC7A5 (also known as LAT1) and SLC3A2 (also known as CD98 heavy chain) form a heteromeric component at the cell surface to facilitate leucine transport[13,14]. SLC3A2 is indispensable for the transport activity and the substrate specificity of SLC7A5[14].

We recently reported that LLGL2 interacts with SLC7A5 and regulates its membrane trafficking, leading us to test the possibility that LLGL2/SCRIB complex may interact with the SLC7A5/SLC3A2 complex to regulate its biology. Consistent with this possibility, SCRIB was present in the anti-SLC7A5 immunoprecipitants along with LLGL2 and SLC3A2, suggesting the formation of a quaternary protein complex (Fig. 2c). To better understand how SCRIB interacts with the SLC7A5-SLC3A2 complex, we performed immunoprecipitation analysis using the anti-SLC7A5 antibody in SLC3A2-knockdown MCF-7 cells. Interestingly, the ability of SLC7A5 to interact with SCRIB, but not LLGL2, was impaired in SLC3A2-KD cells, identifying SLC3A2 as a regulator of SCRIB recruitment into the quaternary complex (Fig. 2d). To understand the role played by SLC3A2 in LLGL2/SCRIB interaction, we expressed Flag-tagged LLGL2 in parental and SLC3A2-KD cells and investigated the ability of LLGL2 to immunoprecipitated endogenous SCRIB. Loss of SLC3A2 impaired the interaction between LLGL2/SCRIB (Fig. 2e), demonstrating an essential role for SLC3A2 in recruiting SCRIB into the ternary complex.

To better understand the interaction between SLC3A2 and SCRIB, we generated N-terminal and C-terminal truncations of SCRIB as HA-tagged proteins and co-expressed them with wild-type SLC3A2 in HEK293T cells (Fig. 2f). Analysis of anti-HA immunoprecipitants demonstrated that SLC3A2 interacts with the N-terminal portion of SCRIB (Fig. 2g). Since we observed co-localization of SCRIB and SLC3A2 at the plasma membrane

(Fig. 2b and Supplementary Fig. 3e), we investigated if membrane localization of SCRIB is required for interaction with SLC3A2 using the SCRIB-P305L mutant[9], which lacks the ability to localize to the cell membrane (Supplementary Fig. 4a–c). Interestingly, SCRIB-P305L and wild-type SCRIB were equally competent for interaction with LLGL2 and SLC3A2 in HEK293T (Supplementary Fig. 4d) and MCF-7 cells (Fig. 2h), demonstrating that membrane localization was not required for the formation of the quaternary complex.

**SCRIB regulates membrane localization of SLC3A2 in ER+ breast cancer cells.** SCRIB and LLGL2 colocalize mainly at the plasma membrane in MCF-7 cells (Supplementary Fig. 5a). Thus, we next addressed how SCRIB and LLGL2 regulate the subcellular distribution of SLC3A2. Neither knockdown of SCRIB or LLGL2 affected the total levels of SLC3A2 in MCF-7 and T47D cells (Fig. 3a, b and Supplementary Fig. 5b–e). Interestingly, SCRIB-KD reduced the membrane-localizing SLC3A2 in both MCF-7 and T47D cells (Fig. 3c and Supplementary Fig. 5f). In contrast, LLGL2-KD did not drastically change the membrane localization SLC3A2 but dramatically impaired membrane localization of SLC7A5 in MCF-7 cells (Fig. 3d). These results suggest that the SCRIB-SLC3A2 and LLGL2-SLC7A5 associate in the cytosol, and the LLGL2-SLC7A5-SLC3A2-SCRIB quaternary complex is shuttled to the plasma membrane. Interestingly, the growth defect of SCRIB-KD cells was partially rescued by the expression of SCRIB-P305L as the expression of wild-type SCRIB (Fig. 3e, f). Considering that SCRIB-P305L mutant interacts with LLGL2 (Fig. 2h), SCRIB works as a scaffold of SLC7A5-SLC3A2 complex to direct the membrane localization of SLC3A2 in ER+ breast cancer cells.

To elucidate the role of SLC3A2 in breast cancer cells, we examine the expression levels of SLC3A2 in breast cancer tissues. SLC3A2 expression was detected in ER+ and ER− breast cancer

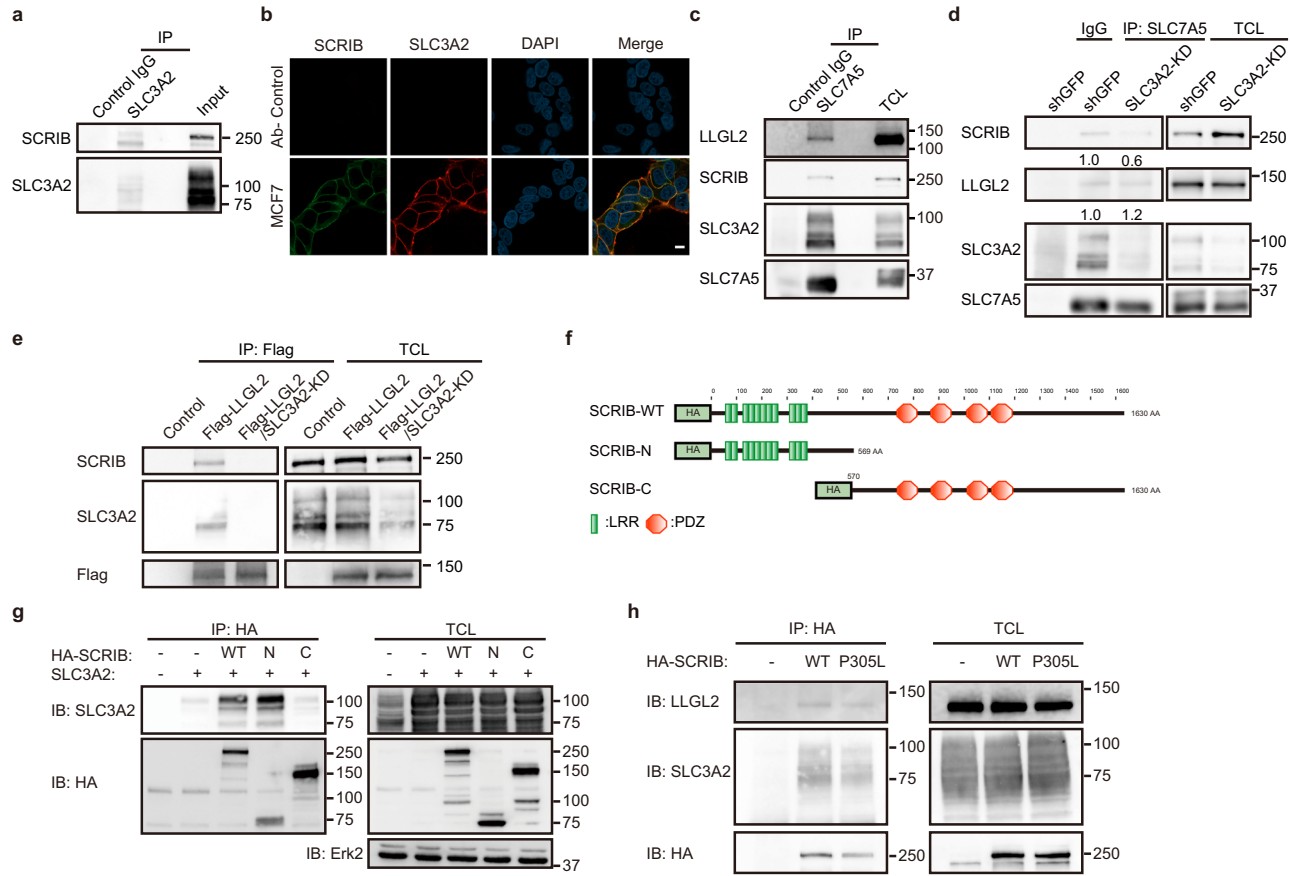

**Fig. 2 SCRIB targets SLC3A2 in ER+ breast cancer cells. a** Detection of endogenous SCRIB-SLC3A2 interaction in MCF-7 cells. **b** Co-localization of SCRIB and SLC3A2 in MCF-7 cells. Scale bar, 10 μm. **c** Immunoprecipitants using the anti-SLC7A5 antibody in MCF-7 cells. **d** Immunoprecipitation of SLC7A5 in SLC3A2-KD cells. **e** Immunoprecipitation of Flag-LLGL2 using anti-Flag antibody in SLC3A2-KD MCF-7 cells. **f** Cartoon representation of wild-type SCRIB and SCRIB mutants. **g** Immunoprecipitation of SCRIB in HEK293T transfected with indicated vectors. **h** Immunoprecipitation of wild-type and mutant SCRIB in MCF-7 cells.

patients' tissue by immunohistochemistry staining (Fig. 4a). SLC3A2 is broadly expressed in ER+, HER2+, and basal breast cancer cell lines without any specific expression pattern (Fig. 4b). Unexpectedly, the cellular surface protein levels of SLC3A2 are high in ER+ breast cancer cells than in ER− breast cancer cells, like the pattern we had previously reported for surface protein levels of SLC7A5[7] (Fig. 4c, d).

Meanwhile, SLC3A2 has been reported to bind with cystine transporter xCT/SLC7A11, which has an essential function in the growth of ER- breast cancer cells[15,16]. Thus, we examined the possibility that SCRIB mediates SLC3A2-SLC7A11 complex formation in ER+ breast cancer cells. To examine whether the complex formation between SLC7A11 and SLC3A2 is mediated by SCRIB, we co-expressed HA-SCRIB, SLC3A2, and Flag-SLC7A11 in HEK293T cells and the cell lysates were immunoprecipitated with anti-HA antibody. Intriguingly, SCRIB forms complex with SLC3A2 and SLC7A11 (Fig. 4e). However, the protein levels of SLC7A11 in ER+ breast cancer cell lines are quite lower than in ER− breast cancer cells (Fig. 4f), suggesting that SCRIB-SLC3A2 complex mainly targets SLC7A5 in ER+ breast cancer cells.

**SLC3A2 promotes cell proliferation by stabilizing SLC7A5 protein in ER+ breast cancer cells.** High expression of *SLC3A2* mRNA correlated with poor clinical prognosis in ER+/PR+ but not ER−/PR− breast cancer patients (Fig. 5a). Consistently,

overexpression of SLC3A2 promoted cell proliferation in both MCF-7 and T47D cells (Fig. 5b, c and Supplementary Fig. 6a, b). Conversely, knockdown of SLC3A2 (SLC3A2-KD) suppressed cell proliferation in ER+ breast cancer cells in both 2D and suspended culture conditions (Fig. 5d–f and Supplementary Fig. 6c–h). Knockdown of SLC3A2 did not affect cell viability in MCF-7 and T47D cells (Supplementary Fig. 6i–l). SLC3A2-KD suppressed cell proliferation in the culture medium supplemented with 10% FBS (Supplementary Fig. 6m). In addition, SLC3A2-KD failed to form tumor in vivo when orthotopically transplanted into immunocompromised mice (Fig. 5g and Supplementary Fig. 6n). These results indicate that SLC3A2 is an essential regulator of cell proliferation in ER+ breast cancer cells.

To understand the relationship between SLC3A2 and SLC7A5 in their ability to regulate the proliferation of ER+ breast cancer cells, we analyzed changes in the protein levels of each other. Unexpectedly, SLC3A2-KD decreased the total protein levels of SLC7A5 in ER+ breast cancer cells (Fig. 5h and Supplementary Fig. 6o), and conversely, SLC7A5-KD reduced total protein levels of SLC3A2 (Fig. 5i and Supplementary Fig. 6p). These results suggest that SLC7A5 and SLC3A2 protein levels are co-regulated, identifying a mechanistic relationship between SLC7A5 and SLC3A2 in ER+ breast cancer cells.

We next investigated the relationship between SLC3A2 and metabolome in ER+ breast cancer cells. SLC3A2-KD cells showed a substantial decrease of intracellular amino acids and a branched-chain amino acid metabolite (Fig. 5j and Supplementary Fig. 6q).

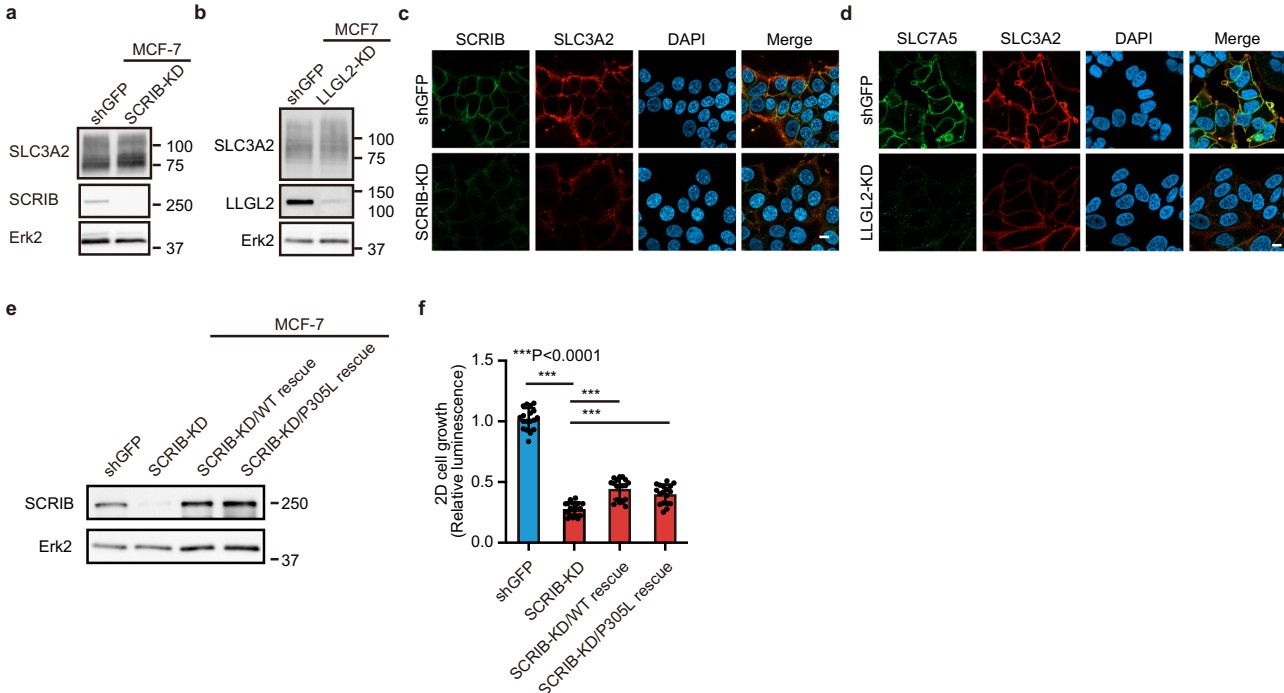

**Fig. 3 SCRIB regulates membrane localization of SLC3A2 in ER+ breast cancer cells. a** Total protein level of SLC3A2 in SCRIB-KD MCF-7 cells. **b** Total protein level of SLC3A2 in LLGL2-KD MCF-7 cells. **c** Immunostaining images of SCRIB and SLC3A2 in MCF-7 cells. Scale bar; 10 μm. **d** Immunostaining images of SLC7A5 and SLC3A2 in LLGL2-KD MCF-7 cells. Scale bar; 10 μm. **e** Rescue of SCRIB expression in SCRIB-KD MCF-7 cells. **f** 2D growth of SCRIB-KD rescued MCF-7 cells. Data in (**f**) are shown as mean ± s.e.m.; n = 18 from six biological replicates performed in triplicate. Statistical analysis was conducted by one-way ANOVA followed by Tukey's posttest.

Thirty-six metabolites were downregulated in SLC3A2-KD in MCF-7 and T47D cells. Comparative metabolite analysis identified eight metabolites, including leucine, that were downregulated in SLC3A2-KD and SCRIB-KD in ER+ breast cancer cells (Fig. 5k, l), suggesting that SCRIB and SLC3A2 targets leucine uptake in ER+ breast cancer cells.

**SCRIB is a target of MYC.** As we observed for LLGL2[7], estrogen (E2)-stimulation-induced SCRIB expression in MCF-7 and T47D cells (Fig. 6a and Supplementary Fig. 7a). However, unlike LLGL2, we did not find the evidence for ER binding site in *SCRIB* promoter/enhancer using a chromatin immunoprecipitation sequencing (ChIP-seq) data from estrogen-stimulated MCF-7 (Gene Expression Omnibus accession number: GSM1669087) and estrogen/progesterone-stimulated T47D cells (GSM1669014) (Supplementary Fig. 7b), suggesting that SCRIB is likely an indirect target of ER.

Interestingly, ChIP-Seq analysis of MYC (GSM808755) and its binding partner MAX (GSM1010863) revealed binding sites on the *SCRIB* gene in MCF-7 cells at the predicted promoter or enhancer regions (Fig. 6b and Supplementary Fig. 7c). The MYC and the MAX-binding sites at the *SCRIB* gene coincided at an open chromatin region characterized by histone H3 acetylation (H3K27ac) at predicted promoter or enhancer regions (Fig. 6b and Supplementary Fig. 7c). Downregulation of MYC by small interference RNA (siRNA) reduced total protein levels of SCRIB in MCF-7 cells (Fig. 6c). Besides, genome editing at the predicted MYC-binding site (chr8:143794647-143794982) and non-functional intronic region in *SCRIB* gene were performed using MCF-7 cells. We successfully eliminated the targeted regions of *SCRIB* gene by CRISPR-Cas9 system (Fig. 6d and Supplementary Fig. 7d, e) and noticed that SCRIB expression is downregulated in the cells that is deleted MYC-binding site in *SCRIB* gene (Fig. 6e). Since MYC expression is strongly induced by ER activation

(Fig. 6a and Supplementary Fig. 7a)[17,18], our results suggest that SCRIB is regulated by the ER-MYC pathway in ER+ breast cancer cells.

**SCRIB and SLC3A2 are required for the growth of tamoxifen-resistant cells.** High SCRIB and SLC3A2 mRNA expression was associated with poor clinical prognosis in ER+ breast cancer patients receiving tamoxifen treatment (Fig. 7a). Total protein levels of SCRIB, SLC3A2, and MYC were markedly upregulated in tamoxifen-resistant cells (Fig. 7b). Downregulation of MYC by siRNA reduced total protein levels of SCRIB in tamoxifen-resistant cells (Fig. 7c). Downregulation of SCRIB or SLC3A2 expression by shRNA in tamoxifen-resistant cells repressed cell proliferation in tamoxifen-resistant cells (Fig. 7d–g), suggesting that SCRIB-SLC3A2 regulates the growth of tamoxifen resistant cells. Furthermore, downregulation of SCRIB or SLC3A2 sensitized tamoxifen-resistant cells to tamoxifen in nutrient stress condition (Fig. 7h), identifying SCRIB-SLC3A2 as regulators of tamoxifen resistance.

## Discussion
Taken together, our results identify a novel pathological role of SCRIB in ER+ breast cancer cells (Fig. 7i). We also report a function for the SCRIB/LLGL polarity complex in cancer cells as a regulator of membrane trafficking of the SLC7A5/SLC3A2 amino acid transporter complex. The SLC7A5/SLC3A2 plays a pivotal role in multiple cancer[15], suggesting that SCRIB/LLGL2 cell polarity complex plays a hitherto unknown cancer-promoting role in multiple cancers. This observation also highlights the ability of cancer cells to repurpose polarity protein complexes involved in normal cell polarity for promoting cancer cell proliferation and drug resistance.

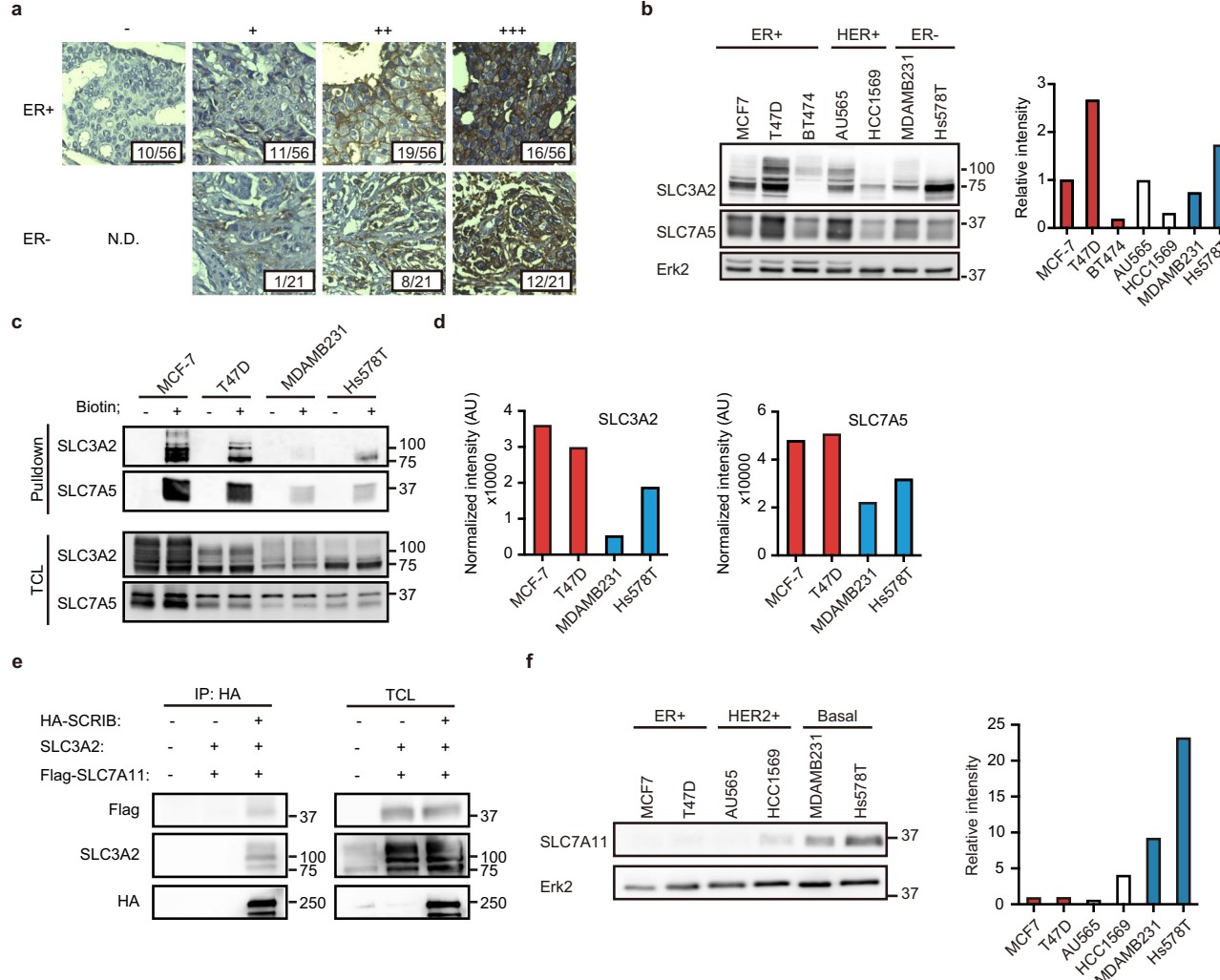

**Fig. 4 SCRIB regulates intracellular localization of SLC3A2 in ER+ breast cancer cells. a** Immunohistological images of SLC3A2 in ER+ and ER− breast cancer. The numbers of specimens were indicated in each image. **b** Total protein levels of SLC3A2 and SLC7A5 in breast cancer cell lines. Immunoblot images (left) and the relative ratio of signal intensities are shown (right). **c** Cellular surface protein levels of SLC3A2 and SLC7A5 in breast cancer cell lines. **d** Normalized intensity of surface protein levels of SLC7A5 and SLC3A2 in ER+ or ER− breast cancer cells. **e** Immunoprecipitation of HA-SCRIB using anti-HA antibody in HEK293T cells. **f** Total protein levels of SLC7A11 in breast cancer cell lines. Immunoblot image (left) and the relative ratio of signal intensities are shown (right).

The growth suppression by SCRIB-KD was rescued by the compensation of SCRIB expression with both wild-type SCRIB and SCRIB P305L. These results suggest that the membrane trafficking of SLC3A2 is regulated by the complex formation with SCRIB in ER+ breast cancer cells. Considering that Discs-large (Dlg) directs the membrane localization of Scrib in fly epithelium during cell polarization[19], DLG and/or LLGL2 might relate to the regulation of membranous SLC3A2 by SCRIB. Besides, our results are consistent with the fact that polarity regulation by SCRIB is tightly link to complex formation with Discs-large (DLG) and LLGL2 although the direct molecular link between cell polarization and membranous SCRIB has not been completely understood.

We report that SCRIB responds to estrogen in MYC-dependent manner. SCRIB levels were not specifically upregulated in ER+ breast cancer. This observation is consistent with the high MYC activity observed in most subtypes of breast cancer. We have previously demonstrated that SCRIB was frequently mislocalized in triple-negative breast cancers (TNBC) and that expression of a membrane localization defective SCRIB mutant

(P305L) activates the Akt pathway to promote cell proliferation in breast cancer cells[9]. Since SCRIB P305L can form the quaternary complex (Fig. 2h) involving SLC7A5/SLC3A2 and wild-type SCRIB can form the complex with SLC7A11 via SLC3A2, it is likely that SCRIB forms a complex with the SLC7A5 and SLC7A11 transporters in a competitive manner, providing a possible explanation for why we did not see an increase in surface levels of SLC7A5 in TNBC cancer[7]. Consistent with the logic, TNBC cells are sensitive to glutamine transporter, xCT/SLC7A11, function[16], suggesting that breast cancer subtypes may have differential sensitivity to amino acid transporters[15]. Further studies would be needed to understand how cell surface levels of transporters such as xCT are regulated in TNBC.

Results presented here along with others[7,20,21] point to an emerging role in amino acid metabolism changes in ER+ cancer and endocrine resistance. The select number of metabolites downregulated in both SLC3A2-KD and SCRIB-KD cells provide a fascinating insight. Amino acids (Leu, Tyr) are substrates of SLC7A5/SLC3A2, and acylcarnitine is generated by CPT1 using acyl-CoA, a product of leucine catabolism. Moreover, cystathionine

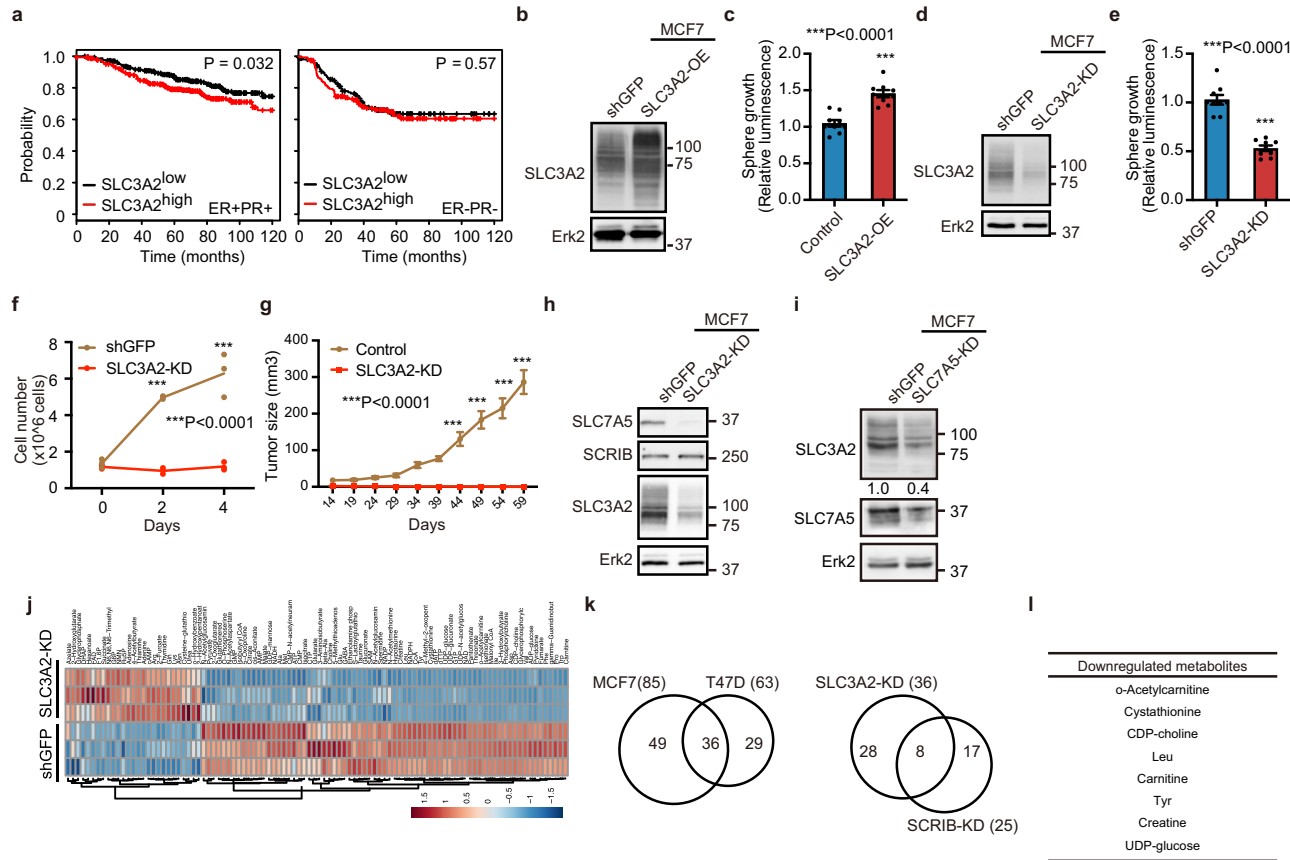

**Fig. 5 SLC3A2 promotes cell proliferation in ER+ breast cancer cells. a** Kaplan–Meier plots of ER+/PR+ and ER−/PR− BC patients' survival.
**b** Overexpression of SLC3A2 in MCF-7 cells. **c** Cell viability assay of SLC3A2-OE MCF-7 cells. **d** Knockdown of SLC3A2 in MCF-7 cells. **e** Cell viability assay
of SLC3A2-KD MCF-7 cells. **f** 2D growth of SLC3A2-KD MCF-7 cells. **g** Tumor growth of SLC3A2-KD MCF-7 cells in NSG mice. **h** Total protein levels of
SLC7A5 in SLC3A2-KD MCF-7 cells. **i** Total protein levels of SLC3A2 in SLC7A5-KD MCF-7 cells. **j** Heatmap of metabolites in SLC3A2-KD MCF-7 cells.
**k** Venn diagram representation of the number of intracellular metabolites downregulated by SLC3A2-KD in MCF-7 and T47D cells (left) and
downregulated in both SLC3A2-KD and SCRIB-KD cells (right). **l** List of metabolites downregulated by SLC3A2-KD and SCRIB-KD. Data **c**, **e**, **f**, and **g** are
shown as mean ± s.e.m.; **c**, and **e**; $n = 9$ from three biological replicates performed in triplicate, **f**; $n = 3$, **g**; $n = 5$ for control cells and $n = 4$ for SLC3A2-KD
cells. Statistical analysis was conducted by $t$-test (**c**, **e**), and two-way ANOVA followed by Tukey's posttest (**f**) or Sidak's posttest (**g**).

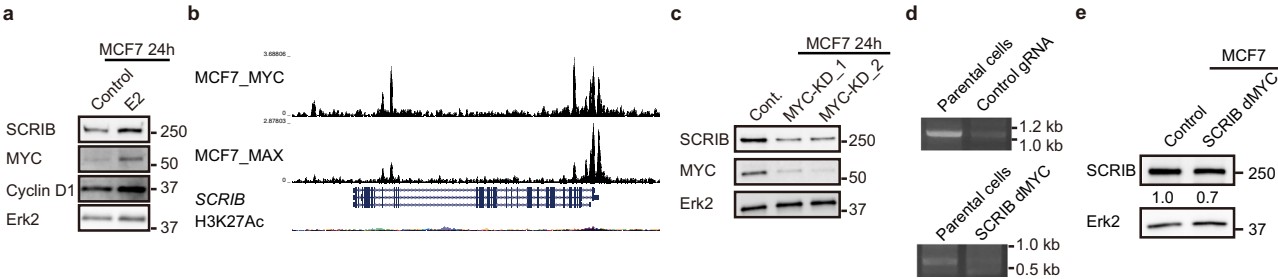

**Fig. 6 SCRIB is a target of MYC. a** SCRIB induction by E2 stimulation in MCF-7 cells. **b** ChIP-Seq analysis of MYC and MAX in MCF-7 cells. **c** The protein
levels of SCRIB in MYC-KD MCF-7 cells. **d** PCR detection of genomic deletion in MCF-7 cells. **e** Total protein levels of SCRIB in genome-edited MCF-7 cells.

is used for cysteine biosynthesis from methionine, suggesting a role
for amino acid metabolisms. How the changes in metabolome
promote proliferation and drug resistance will be an important and
new avenue of investigation.

## Methods

**Cell lines**. MCF-7 cells were cultured in MEM (Nacalai # 21443-15) supplemented
with 10% heat-inactivated FBS, 1 mM sodium pyruvate, and 10 μg/ml insulin.
T47D cells were cultured in RPMI 1640 (Nacalai #30264-85) supplemented with
10% heat-inactivated FBS and 5 μg/ml insulin. AU565 cells, HCC1569 cells, and
BT474 cells were cultured in RPMI 1640 (Nacalai # 30264-85) supplemented with
10% heat-inactivated FBS. HEK293T cells, MDA-MB-231 cells, and Hs578T cells

were cultured in DMEM (High glucose, Wako # 044-29765) supplemented with
10% FBS. Tamoxifen-resistant MCF-7 cells were cultured in phenol-red free RPMI
1640 (Nacalai # 06261-65) supplemented with 10% charcoal-dextran treated FBS
(Cytiva) and 100 nM tamoxifen (Sigma). The parental MCF-7 cells were cultured
in RPMI 1640 supplemented with 10% heat-inactivated FBS. HEK293T, BT474,
AU565, HCC1569, MCF-7, T47D, MDAMB231, and Hs578T cells were derived
from ATCC. Tamoxifen-resistant MCF-7 cell line and the parental MCF-7 cells
were kindly provided by Dr. Rachel Schiff from Baylor College of Medicine. All cell
lines were monitored by MycoAlert PLUS (Lonza) and maintained in
Mycoplasma-free condition.

**Antibodies and reagent**. Anti-LAT1 (#5347), anti-4F2hc (#47213), anti-xCT/
SLC7A11 (#12691) and anti-HA (#3724) were purchased from Cell Signaling. Anti-

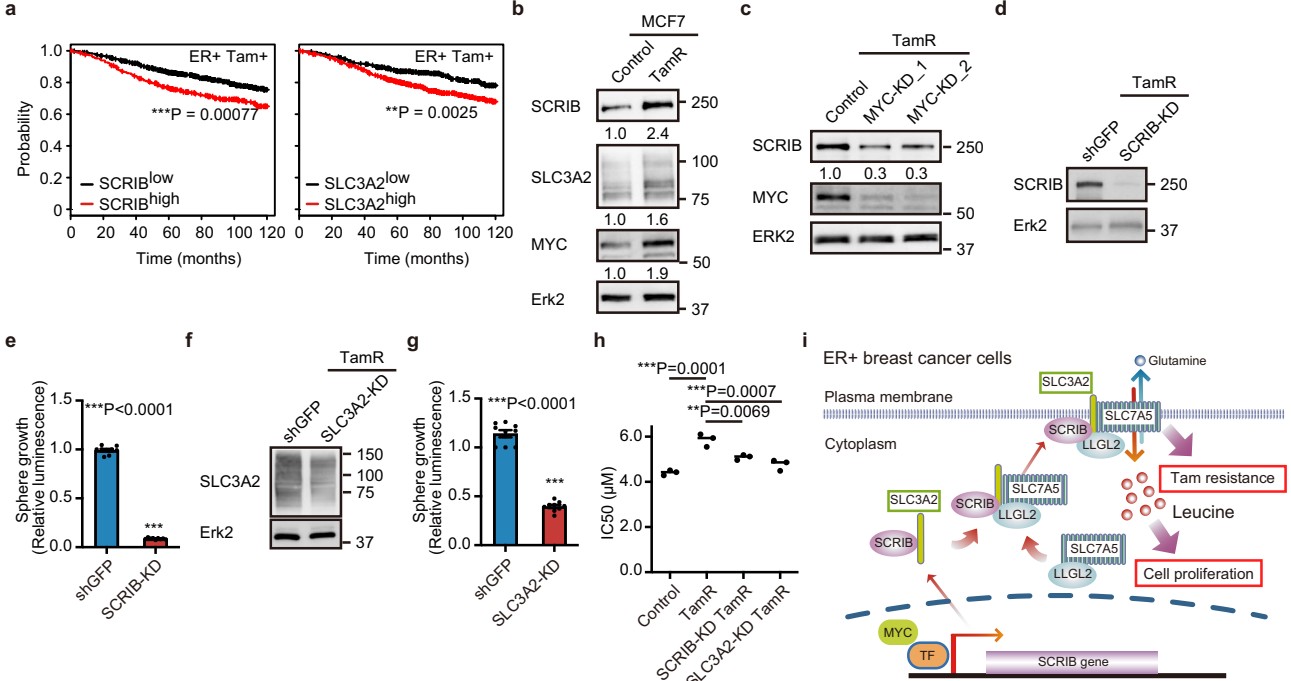

**Fig. 7 SCRIB is required for the growth of tamoxifen-resistant (TamR) breast cancer cells. a** Kaplan–Meier plots of ER+ breast cancer patients treated with tamoxifen. **b** Total protein levels of SCRIB, SLC3A2, and MYC in TamR cells. **c** Knockdown of MYC in TamR cells. **d** Knockdown of SCRIB in TamR cells. **e** Cell viability assay in SCRIB-KD TamR cells. **f** Knockdown of SLC3A2 in TamR cells. **g** Cell viability assay in SLC3A2-KD TamR cells. **h** IC50 values of parental, tamoxifen-resistant, and SCRIB-KD or SLC3A2-KD tamoxifen-reisistant MCF-7 cells in nutrient stress condition. **i** A model of the regulation in leucine uptake by SCRIB-SLC3A2/LLGL2-SLC7A5 pathway. Data **e, g** and **h** are shown as mean ± s.e.m.; **e** and **g**; $n = 9$ from three biological replicates performed in triplicate, **h**; $n = 3$. Statistical analysis was conducted by $t$-test (**e, g**) and one-way ANOVA (**h**).

SCRIB (C-20), anti-SCRIB (C-2), anti-LLGL2 antibody (A-4), anti-CD98 (4F2), anti-CD98 (E-5), anti-Erk2 (C-14), and anti-cyclin D1 (H-295) were purchased from Santa Cruz. Anti-HA, anti-CD98hc (BMP090), anti-SLC7A5 (BMP011), and anti-DDDDK-tag (M185) antibodies were purchased from MBL. Anti-SLC7A5 (HPA052673) and anti-SCRIB (HPA023557) antibodies were purchased from ATLAS ANTIBODIES. Anti-Flag antibody (M2) was purchased from Sigma.

**Immunoblot**. Western blot was conducted as previously described[22]. Briefly, cells were lysed with lysis buffer (50 mM Tris-HCl pH 7.5, 150 mM NaCl, 5 mM EDTA, 1% Triton X-100, protease inhibitors and phosphoSTOP tablets (Roche)). The supernatant was subject to SDS-PAGE, and proteins were transferred onto the PVDF membrane (Millipore #IPVH00010). After blocking with 1% BSA/TBS-0.1% Tween 20, the primary antibody (1:1000 dilution) was reacted overnight at 4 °C. HRP-conjugated secondary antibodies (1:10,000 dilution; GE healthcare) and HRP substrate (Pierce) were used for detection. Images were captured using LAS-4000 mini (Fujifilm). The quantification of signal intensity was conducted using ImageJ64 software.

**Generation of viral vector**. To produce the specific SCRIB knockdown cells, pLKO.1 vectors (shGFP for control, SCRIB-KD and SCRIB-KD_2) were prepared. The sequences are 5′-CAACAAGATGAAGAGCACCAA-3′ for targeting GFP sequences, 5′-GGCGAGACTGTAACTAGTGAT-3′ for SCRIB-KD, 5′-CTGG CCTGTGACTAACTAACT-3′ for SCRIB-KD_2 (TRCN0000004458), 5′-CT AGCTCATACCTGTCTGATTT -3′ for SLC3A2 (TRCN0000043387), and 5′-T TGCTGGTGCCGTGGTCATAA-3′ for SLC3A2_2 (TRCN0000430528). For SCRIB-overexpression, HA-tagged SCRIB gene was sub-cloned into pLJM1 vector (Addgene #19319) at AgeI/EcoRI site. Each lentivirus was produced by transfection with pCMV delta R8.2 (Addgene #12263) and pCMV-VSV-G (Addgene #8454) by calcium phosphate transfection in HEK293T cells.

**2D cell proliferation assay**. One million cells were plated on a 6 cm dish. After 24 h culture, cells were trypsinized, and the cell number was counted. This cell number was set as Day 0. After rinse with pre-warmed PBS, culture media were changed to serum-free media (DMEM/F12 with B27 supplement (Life technologies #12587-010) and 20 ng/ml EGF (Peprotech #AF-100-15)). On the indicated day, the number of cells was counted with 0.4% trypan blue staining. The results confirmed with more than two independent experiments with three replicates.

**Cell viability assay**. Cells were plated in ultra-low attachment 96 well plate (Corning) at 10,000 cells per well in DMEM/F12 supplemented with 1x B27 minus vitamin A and 20 ng/ml EGF. On day 4, the cell viability assay was conducted using CellTiter-Glo 3D (Promega) following the instruction. The luminescence intensity was measured by a plate reader (TECAN infinite M200).

**BioID analysis**. Before cell collection, biotin (final concentration of 50 μM) was added to the culture medium. After 24 h labeling, cells were collected and lysed with RIPA buffer (50 mM Tris-HCl pH 7.5, 150 mM NaCl, 1 mM EDTA, 0.1% SDS, 1% Triton X-100, 0.5% sodium deoxycholate) supplemented with phospho-STOP tablet (Roche) and cOmplete (Roche). Two mg of protein lysates were incubated with 20 μl of Streptavidin Sepharose High Performance Beads (GE Healthcare) and rotated for 1.5 h at 4 °C. Streptavidin beads were washed eight times with lysis buffer, and 35 μl of 1X sample buffer was mixed with beads. The purified proteins were eluted by boiling for 10 min.

**Mass spectrometry analysis for BioID**. The eluted samples were reduced with 10 mM Tris (2-carboxyethyl) phosphine hydrochloride (TCEP) at 100 °C for 10 min. Following alkylation with 50 mM iodoacetamide at ambient temperature for 45 min, protein samples were subjected to SDS-PAGE. The electrophoresis was stopped at the migration distance of 2 mm from the top edge of the separation gel. After CBB-staining, protein bands were excised, destained, and cut finely prior to in-gel digestion with Trypsin/Lys-C Mix (Promega) at 37 °C for 12 h. The resulting peptides were extracted from gel fragments and analyzed with Orbitrap Fusion Lumos mass spectrometer (Thermo Scientific) combined with UltiMate 3000 RSLC nano-flow HPLC (Thermo Scientific). Peptides were enriched with μ-Precolumn (0.3 mm i.d. × 5 mm, 5 μm, Thermo Scientific) and separated on AURORA column (0.075 mm i.d. × 250 mm, 1.6 μm, Ion Opticks Pty Ltd, Australia) using the two-step gradient; 2–40% acetonitrile for 110 min, followed by 40–95% acetonitrile for 5 min in the presence of 0.1% formic acid. The analytical parameters of Orbitrap Fusion Lumos were set as follows; Resolution of full scans = 50,000, Scan range (m/z) = 350–1500, Maximum injection time of full scans = 50 ms, AGC target of full scans = 4 × 10^5, Dynamic exclusion duration = 30 s, Cycle time of data-dependent MS/MS acquisition = 2 s, Activation type = HCD, Detector of MS/MS = Ion trap, Maximum injection time of MS/MS = 35 msec, AGC target of MS/MS = 1 × 10^4. The MS/MS spectra were searched against the Homo sapiens protein sequence database in SwissProt using Proteome Discoverer 2.4 software (Thermo Scientific), in which peptide identification filters were set at "false discovery rate < 1%". Label-free relative quantification analysis for identified proteins was performed with the default parameters of Minora Feature Detector node, Feature Mapper node, and Precursor Ions Quantifier node in Proteome Discoverer 2.4 software.

**Biotin-labeling surface proteins and pull-down assay**. Cells were rinsed with ice-cold PBS, and surface proteins were labeled with 400 µM EZ-link-sulfo-NHS-SS-Biotin/PBS for 30 min at 4 °C with gentle rocking. The reaction was quenched with 2 ml of 150 mM glycine. After twice rinse with ice-cold PBS, cells were lysed in lysis buffer (50 mM Tris (pH 7.4), 100 mM NaCl, 5 mM EDTA (pH 7.4), 1% Triton-X-100, 5 mM NaF, plus protease inhibitors and phosphoSTOP tablets (Roche)). 50 µg of protein lysate was incubated with 20 µl of Streptavidin Sepharose High Performance Beads (GE Healthcare) and rotated for 1.5 h at 4 °C. Streptavidin beads were washed three times with lysis buffer, and proteins were eluted with 1X sample buffer.

**Immunoprecipitation**. For detecting SCRIB-SLC3A2 interaction, cells were cultured in DMEM/F12 media supplemented with B27 minus vitamin A and 20 ng/ml of EGF for 2 days before harvest. Cells were collected and lysed with lysis buffer (50 mM Tris pH7.4, 100 mM NaCl, 5 mM EDTA pH7.4, 1% Brij-35, 2 mM sodium orthovanadate, 1 mM NaF and 1 mM beta-glycerophosphate, protease inhibitor and phosphoSTOP tablets (Roche)). Cell lysates were pre-cleared with 20 µl of Protein G Sepharose beads (GE Healthcare, #17-0168-01) for 10 min, and the cleared cell lysates were transferred to a new tube. Four µl of anti-SLC3A2 antibody, 4 µl of anti-SLC7A5 (BMP011), 4 µl of anti-DDDDK-tag (Flag-tag) or 4 µl of anti-HA antibody (Cell Signaling #3724 or MBL Life science #M180-3) was added into cell lysates and incubated for 1 h at 4 °C with rotation. SCRIB or SLC3A2 complexes were pulled down with 20 µl of Protein G Sepharose beads (GE Healthcare, #17-0168-01) for 45 min. After 3 to 5 times washing with lysis buffer, precipitants were eluted with 1X SDS sample buffer and subject to western blot. For the investigation of the SCRIB interacting site with SLC3A2, HEK293T cells were transfected with indicated vectors. After 48 h from transfection, immunoprecipitation was performed as described above.

**Orthotropic tumor xenograft**. The six weeks old female mice (NOD.Cg-Prkdcscid Il2rgtm1Wjl/SzJ) were purchased from Charles River. 90-day slow-release estrogen pellets were subcutaneously implanted into 8-week-old mice two days before cell injection (0.72 mg, Innovative Research of America). Cells suspension (1 × 10⁵ cells) in 50 µl of DMEM/F12 with 20 ng/ml EGF and 1X B27 minus vitamin A were mixed with 50 µl of Matrigel. The 100 µl of cell mixture was injected into the 4th mammary fat pad using a 27 G 1/2 inch 1 ml syringe as described[23]. The tumor size was measured every three days from day 15 (for SCRIB-KD cells) or every five days from Day 14 (for SLC3A2-KD cells). Tumor volumes were estimated with the formula: volume = $(2a \times b)/2$, where $a$ = shortest and $b$ = longest tumor lengths, respectively, in millimeters. The two-way ANOVA and Sidak's multiple comparison tests were conducted using Prism 9 software. All mouse experiments had the approval of the Laboratory Animal Center, Keio University School of Medicine (Protocol #19046-(0)) and were carried out following the 'Guide for the Care and Use of Laboratory Animals'.

**Immunohistochemistry staining**. Tissue array slides, in which ER status was qualified, were purchased from USBiomax (Catalog #: BC081120c). Slides were baked at 60 °C for 30 min. After deparaffinization, antigen retrieval was conducted for 15 min using a pressure boiler in 10 mM citrate buffer containing 0.05% Tween-20 (pH6.0). After cooling down to room temperature, endogenous HRP was inactivated with 3% H₂O₂ for 10 min. From blocking procedures, VECTASTAIN UNIVERSAL Elite ABC KIT (Vector Laboratories; PK-6200) and ImmPACT DAB Peroxidase Substrate Kit (Vector Laboratories; SK-4105) were used following kit protocol. The tissue slides were counterstained with hematoxylin solution (Sigma; HHS16-500ML). After slides mount with Mount-Quick (Daido-Sangyo; DM-01), tissue images were captured using a microscope (KEYENCE BZ-X700).

**Metabolome analysis**. Metabolome analysis was conducted as described in the previous report[11]. Briefly, one million cells were cultured in a poly-L-lysine-coated 6 cm diameter dish (Iwaki #4010-040) with an extra culture dish for cell counting. After 24 h, the cell culture medium was replaced with DMEM/F12 with 20 ng/ml EGF and 1X B27 minus vitamin A. Before 2 h from the metabolite extraction, cell culture medium was replaced with DMEM/F12 with 20 ng/ml EGF and 1X B27 minus vitamin A. Cells were rinsed with 5% mannitol solution, and cells were lysed with 700 µl of methanol containing 25 µM internal standards[11]. For the normalization of metabolite concentrations, cell number was counted. For the analysis of tumor tissues, 30 mg of tumor was lysed with 700 µl of methanol containing internal standards. Metabolome analysis was performed by capillary electrophoresis time-of-flight mass spectrometry (CE-TOFMS) using Agilent 7100 CE capillary electrophoresis (Agilent Technologies). System control and data acquisition were performed by using Agilent MassHunter Workstation, and data analysis was conducted using Keio MasterHands software. The concentration of each metabolite was calculated by normalizing with cell number as previously described[11,24,25].

**ChIP-seq data analysis**. ChIP-seq data for ESR1 and MYC in MCF7 cells were analyzed from the Cistrome database (http://www.cistrome.org/Cistrome/Cistrome_Project.html) as previously described[26].

**CRISPR-Cas9-mediated deletion of MYC-binding domain**. The genomic regions of SCRIB (intron 24 and intron 27) were used to design guide RNA (gRNA) using the Cistorome Database (http://www.cistrome.org/SSC). Two gRNAs were selected to delete MYC-binding site (chr8:143794647-143794982) at intron 24 and two gRNAs were selected for control which delete 555 bp in intron 27 region in the SCRIB gene. Targeting vectors were prepared as previously described[7].

**Genomic PCR**. Genomic PCR was performed to confirm the targeted genomic deletion in SCRIB gene. The sequences of each guide RNA and PCR primer are given in Supplementary Fig. 7. Genomic DNAs were extracted using Monarch Genomic DNA Purification Kit (NEB T3010S). We used 10 ng genomic DNA for the PCR template. PCR was conducted using KOD Plus Neo (TOYOBO KOD-401) according to the manufacturer's protocol. Thermal steps were 94 °C for 2 min, and 35 cycles of 98 °C for 10 s, 60 °C for 30 s, and 68 °C for 1 min. PCR products were subjected to agarose gel electrophoresis.

**IC50 calculation**. Cells were plated into 96 well-plates at the cell density of 7000 cells per well. After 24 h culture, the culture medium was replaced with the assay medium (DMEM/F12 supplemented with B27 and 20 ng/ml EGF) containing a series of tamoxifen dilutions at final concentration of 0 nM, 100 nM, 500 nM, 750 nM, 1 µM, 2.5 µM, 5 µM, 7.5 µM, 10 µM, and 50 µM. On day 4, the cell viability assay was conducted using CellTiter-Glo 3D (Promega) following the instruction. The luminescence intensity was measured by a plate reader (TECAN infinite M200). The values of IC50 were calculated using Prism 9 software.

**Statistics and reproducibility**. All statistical analyses were conducted using Prism 9 (GraphPad Software). Statistical significances were shown as n.s. (not significant), $*P < 0.05$, $**P < 0.01$, $***P < 0.001$. The number of experiments was noted in figure legends. The sample size used in each experiment was not predetermined or formally justified for statistical power. To assess the statistical significance of a difference between two treatments, we used two-tailed Student's $t$-tests. To assess the statistical significance of differences between more than two treatments, we used one-way or two-way ANOVA following Tukey's multiple comparison test or Sidak's multiple comparison test.

**Reporting summary**. Further information on research design is available in the Nature Research Reporting Summary linked to this article.

## Data availability

The Kaplan–Meier plot data that supports this study's findings are available in Kaplan–Meier Plotter (http://kmplot.com/analysis/) with the identifier (10.18632/oncotarget.10337). The ChIP-seq data "GSM1534722"[27], "GSM808755"[28], "GSM1010863"[29], and "GSM1669014"[27] that support the findings of this study are available from Cistrome Data Browser (http://cistrome.org/db/#/). Uncropped images of blots/gels are shown in Supplementary Figure. Source Data for Figs. 1–5 are available in Supplementary Data 1. Source Data for Supplementary Figs. 1–7 are available in Supplementary Data 2. All other relevant data are available from the corresponding author on reasonable request.

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

## Acknowledgements

We thank members of the Soga laboratory for the discussion. Funding supported this work from a Grant-in-Aid for Scientific Research (C) and Grant-in-Aid for Research Activity Start-up from the Japan Society for the Promotion of Science (JSPS), The Mochida Memorial Foundation, The Yasuda Medical Foundation, Astellas Foundation for Research on Metabolic Disorders, Kanae Foundation, The Uehara Memorial Foundation, and The Naito Foundation (Y.S.) and the foundation from Yamagata prefectural government and the City of Tsuruoka (Y.S., S.M., K.E., S.A., M.O., A.U., M.T., and T.S.). S.K.M was supported by a grant from the Breast Cancer Research Foundation.

## Author contributions

Y.S., M.T., T.S., and S.K.M. designed, performed, and interpreted experiments and co-wrote the paper. Y.S. and S.M. carried out all the experiments. N.O. and K.U. performed BioID analysis. K.E., S.A., M.O., and A.U. performed metabolome analysis.

## Competing interests

The authors declare no competing interests.
