## [Peer Review File · Communications Biology]

Reviewers' comments:

Reviewer #1 (Remarks to the Author):

Saito and colleagues present an elegant analysis of protein-protein interactions for amino acid transporters and the polarity protein SCRIB in ER+ breast cancer cells. The area of ER+ breast cancer therapy resistance is of high impact and clinical importance. To directly test the role of the SCRIB complex in tamoxifen resistance, I have suggested experiments below but I would not say that important question has been addressed in the current manuscript. The title stating this regulates tamoxifen resistance is an overstatement as this has not been experimentally tested.

Specific comments:

1. There are typographical/grammar/syntax errors throughout the manuscript that should be corrected
2. Is SCRIB mRNA correlated with survival in other types of breast cancer or is this specific to ER+? Is it dependent on PR status? How does this correlation look in TCGA HER2 and TNBC samples?
3. Fig 1f- Since SCRIB KD causes such drastic cell growth inhibition in vitro, it seems likely upon xenotransplantation into mammary glands, the cells never really begin forming a tumor in the first place. This would preclude our ability to judge whether tumor growth in vivo is affected. Rather, it might be that tumor initiating capacity is impacted rather than only proliferation. Authors should address this discrepancy.
4. Regarding figure 1 and S1- Can authors please clarify whether the growth inhibitory effect of SCRIB KD is specifically observed in suspension/spheroid culture or is this also seen in adherent cultures? How about low-nutrient vs regular media?
5. Figure 1g metabolite experiment-
 - a. Can authors please clarify how they analyzed metabolites from KD tumors? The plot in 1f suggests there is no observable tumor to harvest. Did tumors grow at all in this experiment?
 - b. I am concerned about the loss of metabolites (were any metabolites enriched in KD cells?) being simply a result of drastically reduced cell growth/viability.
6. The sphere growth inhibition induced by SCRIB KD seems very different when comparing figure 1c and 1i – what could be the reason for this difference?
7. Can authors explain what the various bands clearly visible on western blots for SLC3A2 and SLC7A5 are? Are multiple isoforms expected?
8. I have the same comment for SLC3A2 KD (fig 4e-g) as above written in point #3
9. Regarding figure 4k, I have the same concerns as written above in point #5
10. Figure 5- I am concerned the reduction in SCRIB levels following MYC KD could be a secondary effect of the drastic growth inhibitory effect MYC KD is likely to have. Instead, authors should test whether mutation/deletion of the MYC binding site in the SCRIB promoter/enhancer disrupts SCRIB protein expression
11. Final results section of text- please remove the statement in the first sentence' suggesting a specific role for SCRIB and SLC3a2 in management of ER+ breast cancer. Authors should also evaluate whether the survival relationship with SCRIB and SLC3A2 is observed in ER+ patients who had NOT been treated with tamoxifen.
12. Figure 5- the impacts of SCRIB Kd on cell proliferation/viability seem very similar in MCF7/T47D and the TamR cells. I do not agree that authors can conclude that SCRIB mediates Tam resistance from these data.
13. Throughout, authors should test whether SCRIB and SLC3A2 impact cell proliferation rate, and/or apoptosis- any combination of these could explain the relative ATP viability assay results presented.
14. Authors should at least discuss, and ideally experimentally test, whether the polarity regulating role of SCRIB is related to its impact on cell growth and/or leucine import and/or tamoxifen resistance.
15. Figure 5- from this figure we cannot conclude whether SCRIB is important for tamoxifen

resistance, as the title of this section implies. To test this, authors need to measure cell viability / tamoxifen IC50 in the presence and absence of SCRIB KD.

Reviewer #2 (Remarks to the Author):

The manuscript submitted by Saito et al. describes a role for SCRIB in regulating amino acid metabolism in ER+ breast cancer cells. The study is a follow-up to the group's recent finding that LLGL2 impacts cell proliferation under nutrient stress and regulates tamoxifen resistance. Overall this is a novel study that adds to our understanding of mechanisms underlying metabolic functions in breast cancer growth, with a strong link to the cell polarity machinery. The data presented are generally convincing and clear.

The authors demonstrate that knockdown of SCRIB affects global metabolic profiles and associated with SLC3A2 via the amino-terminus, and forms a quaternary complex with LLGL2 and SLC7A5. Is SCRIB/SLC3A2 associated specifically with LLGL2 and SLC7A5, or are there other transporter complexes that SCRIB associates with? For example, SLC3A2 also associates with SLC7A11 (PMID: 16103098). The dramatic changes in metabolism with SCRIB-KD (i.e. almost all metabolic changes are associated with metabolite down-regulation in SCRIB-KD cells) suggest that this may be the case.

The finding that SCRIB expression is regulated by MYC and implicated in tamoxifen resistance is of interest. While SCRIB is upregulated in TamR cells and SCRIB-KD and SLC3A2-KD cells have impaired growth, the conclusion that SCRIB-SLC3A2 are regulators of tamoxifen resistance is an over-interpretation of this data. SCRIB-KD has the same effect in TamR as in normal MCF7 cells. This could be addressed by determining if SCRIB and SLC3A2 knockdown re-sensitize cells to tamoxifen.

I am enthusiastic about this manuscript, however these and additional points need to be addressed.

Specific points:

1. Figures 1a, 1e, 4a, 4b, – what does Probability on the y-axis represent? Overall survival, recurrence-free survival? Other?
2. Figures 1c, 4d, 4e - What does each dot represent? Are these independent experiments, or different wells from the same experiment? This information should be provided in the figure legends.
3. All of the graphs for sphere growth (Fig. 1c, e,) use relative ATP as the measurement. My concern is that this is a metabolic readout for growth. ATP-levels are an indirect measurement of growth and are affected by SCRIB-KD (Supp. Fig 2). Therefore, can an independent measurement of sphere growth should be used to validate the results obtained with the ATP assay? At least for some of the data.
4. Some of the SLC3A2 blots have a large number of bands or smears and are difficult to interpret (e.g. Fig. 2g, TCL; 2h TCL; 5h).
5. Fig. 5f – the blot should include a blot for myc. I expect it would be upregulated in addition to SCRIB. Also, does MYC knockdown in TamR MCF7 cells block upregulation of SCRIB protein?
6. How was the metabolomics data normalized? I did not see this information in the figure legend or methods section. Also, what are the criteria/cut-offs used for determining down-regulation?

Referees' comments:

Reviewer #1:

Saito and colleagues present an elegant analysis of protein-protein interactions for amino acid transporters and the polarity protein SCRIB in ER+ breast cancer cells. The area of ER+ breast cancer therapy resistance is of high impact and clinical importance. To directly test the role of the SCRIB complex in tamoxifen resistance, I have suggested experiments below, but I would not say that important question has been addressed in the current manuscript. The title stating this regulates tamoxifen resistance is an overstatement as this has not been experimentally tested.

- We thank this reviewer for constructive comments on our manuscript. We have included new data proving the evidence that SCRIB and SLC3A2 plays an important role on the regulation of tamoxifen-resistance in ER+ breast cancer cells. The downregulation of SCRIB or SLC3A2 suppressed cell proliferation in tamoxifen-resistant cells, indicating that both SCRIB and SLC3A2 are required for the growth of tamoxifen-resistant cells. Furthermore, we showed that suppression of SCRIB or SLC3A2 expression in TamR cells sensitized to tamoxifen, supporting that SCRIB-SLC3A2 pathway contributes to the molecular mechanisms of the resistant acquisition. We commented back one by one in the following specific comments.

Specific comments:

1. There are typographical/grammar/syntax errors throughout the manuscript that should be corrected.

- We have corrected the most of errors in the manuscript.

2. Is SCRIB mRNA correlated with survival in other types of breast cancer or is this specific to ER+? Is it dependent on PR status? How does this correlation look in in TCGA HER2 and TNBC samples?

- In response to these questions, we performed the correlation analysis in ER+/PR- and ER-/PR+ cohorts (Supplementary Fig. 1b, c). We found that the ER+/PR- or ER-/PR+ cohorts did not show statistical significance, but the dataset does not have enough samples number to draw meaningful conclusions. A more extensive study may be needed to investigate the clinicopathological relationships between single hormone receptor positive breast cancers, SCRIB, and clinical outcome. Because of these results and in response to this reviewer's concerns, we now have taken extreme care by restricting our interpretations to only the ER/PR double positive cohorts as presented in the original manuscript not drawing any attention to patient cohorts with of ER+ alone or PR+ alone cancers.
- Also, we performed same analysis in HER2 and TNBC samples and have included those data in Supplementary Fig. 1d-f. The results showed that the expression levels of SCRIB mRNA did not correlate with poor clinical survival in the group of HER2+ or TNBC

patients (p. 5, line 6-8).

3. Fig 1f- Since SCRIB KD causes such drastic cell growth inhibition *in vitro*, it seems likely upon xenotransplantation into mammary glands, the cells never really begin forming a tumor in the first place. This would preclude our ability to judge whether tumor growth *in vivo* is affected. Rather, it might be that tumor initiating capacity is impacted rather than only proliferation. Authors should address this discrepancy.

- We understand this reviewer's concern. In response, we have showed the magnified graph drawing of tumor growth in Supplementary Fig. 1s. As shown in the graph, SCRIB-KD cells formed tumors and those tumors slowly grew *in vivo*. This result strongly supports that SCRIB-KD suppressed tumor growth with little effects on the tumor initiating capacity of MCF-7 cells.

4. Regarding figure 1 and S1- Can authors please clarify whether the growth inhibitory effect of SCRIB-KD is specifically observed in suspension/spheroid culture or is this also seen in adherent cultures? How about low-nutrient vs regular media?

- This is an important question. In response to this concern, we have included the data on 2D growth of SCRIB-KD cells in Supplementary Fig. 1i and j in addition to the growth data under suspension culture condition (Fig. 1c, e). Consistently, SCRIB-KD suppressed cell proliferation in both suspension and 2D adherent culture condition.

- In addition, we have examined the cell proliferation of SCRIB-KD cells in regular culture condition. The downregulation of SCRIB lead to the growth inhibition in normal culture medium (Supplementary Fig. 1x) whereas LLGL2-KD suppressed cell proliferation only in nutrient stress condition, not in regular culture condition (Saito et al., Nature 2019), suggesting the multiple functions of SCRIB in the growth of ER+ breast cancer cells.

5. Figure 1g metabolite experiment

a. Can authors please clarify how they analyzed metabolites from KD tumors? The plot in 1f suggests there is no observable tumor to harvest. Did tumors grow at all in this experiment?

- In response to this concern, we have included the sentence on metabolites analysis in methods section (p. 23, line 10-12). As argued in the comment #3 section, SCRIB-KD cells formed small tumors. Thus, we have analyzed the metabolites using 30 mg of tumor per samples for metabolome analysis.

b. I am concerned about the loss of metabolites (were any metabolites enriched in KD cells?) being simply a result of drastically reduced cell growth/viability.

- We observed some enriched metabolites such as dTTP, UDP-glucuronate, Asp, NAD⁺, F6P and acetylcholine in SCRIB-KD MCF-7 cells. We agree that some of metabolites may be downregulated by the result of reduced cell growth. However, the growth defects of SCRIB-KD cells were rescued by the treatment of high dose leucine medium. This result

strongly supports that SCRIB regulates leucine uptake in MCF-7 cells. In response to this reviewer's concern, we now have taken extreme care by restricting our interpretations on metabolome analysis to only the leucine downregulation.

6. The sphere growth inhibition induced by SCRIB KD seems very different when comparing figure 1c and 1i – what could be the reason for this difference?

- In response to this reviewer's concern, we have showed the reproduceable data in Fig. 1i. We thought that the difference in the magnitude of growth phenotype has been caused by the knockdown efficiency of SCRIB. The newly prepared SCRIB-KD cells showed growth suppression with same magnitude as shown in other figures, and the growth suppression in SCRIB-KD cells was rescued by the treatment with 10x LQ medium without any change in the growth of control cells.

7. Can authors explain what the various bands clearly visible on western blots for SLC3A2 and SLC7A5 are? Are multiple isoforms expected?

- In response to this concern, we have included the figure that explain the multiple bands in SLC3A2 blot image in Supplementary Fig. 3d. We transfected SLC3A2 cDNA in HEK293T cells and the cell lysates were analyzed by immunoblot using anti-SLC3A2 antibody. As previously reported in Fort et al. (J. Biochem. 282; 31444-31452, 2007; doi:10.1074/jbc.m704524200), immunoblot image of SLC3A2 shows multiple bands due to the posttranslational modification of SLC3A2 such as glycosylation. Monomer (glycosylated form and non-glycosylated form) and proteolytic fragment are indicated in Supplementary Fig. 3d.

8. I have the same comment for SLC3A2 KD (fig 4e-g) as above written in point #3

(Point #3. Fig 1f- Since SCRIB KD causes such drastic cell growth inhibition in vitro, it seems likely upon xenotransplantation into mammary glands, the cells never really begin forming a tumor in the first place. This would preclude our ability to judge whether tumor growth in vivo is affected. Rather, it might be that tumor initiating capacity is impacted rather than only proliferation. Authors should address this discrepancy.)

- In response to this concern, we have included the magnified graph in Supplementary Fig. 6n. In contrast to SCRIB-KD cells, SLC3A2-KD cells did not form tumor as pointed out from this reviewer. In response to this reviewer's concern, we have edited the sentence that "SLC3A2-KD failed to form tumor in vivo when orthotopically transplanted into immunocompromised mice". (p. 11, line 9)

9. Regarding figure 4k, I have the same concerns as written above in point #5

(Point #5. Figure 1g metabolite experiment

a. Can authors please clarify how they analyzed metabolites from KD tumors? The plot in 1f

suggests there is no observable tumor to harvest. Did tumors grow at all in this experiment?

- As commented in previous comment (#8), SLC3A2-KD cells did not form tumor *in vivo*. Therefore, the metabolome analysis data shown in Fig. 4j, 4k and 4l do not include the data derived from the SLC3A2-KD tumor in the original version of manuscript.

b. I am concerned about the loss of metabolites (were any metabolites enriched in KD cells?) being simply a result of drastically reduced cell growth/viability.

- Although the significant growth suppression by SLC3A2 downregulation, we observed the upregulated metabolites in SLC3A2-KD cells (Fig. 4j and Supplementary Figure 6q), suggesting that downregulation of metabolites in SLC3A2-KD cells is not a result of drastically reduced cell growth/viability.

10. Figure 5- I am concerned the reduction in SCRIB levels following MYC KD could be a secondary effect of the drastic growth inhibitory effect MYC KD is likely to have. Instead, authors should test whether mutation/deletion of the MYC binding site in the SCRIB promoter/enhancer disrupts SCRIB protein expression

- In response to this concern, we performed genome editing to modify MYC-binding site in the enhancer regions of SCRIB to address MYC-dependent SCRIB expression. We designed two combinations of guide RNA for CRISPR-Cas9 based genome editing. One pair of gRNA targets non-functional intron of SCRIB gene as a control and the other pair of gRNA targets one of MYC-binding site. As shown in Fig. 5d and 5e, both pairs of gRNA could eliminate the targeted genomic region and the expression of SCRIB was partially reduced by the deletion of MYC-binding site in the SCRIB gene, suggesting that MYC directly targets SCRIB expression in MCF-7 cells.

11. Final results section of text- please remove the statement in the first sentence' suggesting a specific role for SCRIB and SLC3A2 in management of ER+ breast cancer. Authors should also evaluate whether the survival relationship with SCRIB and SLC3A2 is observed in ER+ patients who had NOT been treated with tamoxifen.

- We have removed the sentence as suggested by this reviewer.

12. Figure 5- the impacts of SCRIB KD on cell proliferation/viability seem very similar in MCF7/T47D and the TamR cells. I do not agree that authors can conclude that SCRIB mediates Tam resistance from these data.

- In response to this concern, we examined the IC50 values of parental, tamoxifen-resistant and SCRIB-KD tamoxifen-resistant cells under nutrient stress condition. As shown in Fig. 5m, the SCRIB-KD tamoxifen-resistant cells were sensitized to tamoxifen until similar levels to control cells, suggesting that SCRIB mediates tamoxifen resistance in MCF-7 cells.

13. Throughout, authors should test whether SCRIB and SLC3A2 impact cell proliferation rate, and/or apoptosis- any combination of these could explain the relative ATP viability assay results

presented.

- In response to this concern, we examined cell proliferation by counting alive and dead cell number with trypan blue staining. As shown in Supplementary Fig. 1k, 1l, 1m, 1n, 6i, 6j, 6k and 6l, both SCRIB-KD and SLC3A2-KD suppressed cell proliferation with no effect on the number of dead cells. These results are consistent with the results of relative ATP viability. Therefore, we concluded that relative ATP viability is equivalent to the number of growing cells.

- In addition, to avoid the misleading of growth data, the title of y-axis shown as “relative ATP” is changed to “relative luminescence”.

14. Authors should at least discuss, and ideally experimentally test, whether the polarity regulating role of SCRIB is related to its impact on cell growth and/or leucine import and/or tamoxifen resistance.

- We understand this reviewer’s concern. In response to this comment, we performed rescue experiments in the cell growth assay of SCRIB-KD cells. Unexpectedly, the growth suppression by SCRIB-KD was rescued by the re-expression of both wild-type SCRIB and SCRIB-P305L mutant. This result suggests that the complex formation of SCRIB with SLC3A2 is needed for the membrane localization of SLC3A2, which is important for the leucine uptake via SLC7A5. Besides, this result is consistent with the fact that polarity regulation by SCRIB is tightly link to complex formation with Discs-large (DLG) and LLGL2 although the direct molecular link between cell polarization and membranous SCRIB has not been completely understood (PMID: 32414916). We have added the sentences about this topic in the discussion section (page 14 line 7-15).

- In addition, both SCRIB and SLC3A2 regulate the tamoxifen-resistance phenotypes, implicating the molecular link between cell polarity and tamoxifen resistance although further investigation is absolutely needed to confirm this idea.

15. Figure 5- from this figure we cannot conclude whether SCRIB is important for tamoxifen resistance, as the title of this section implies. To test this, authors need to measure cell viability / tamoxifen IC50 in the presence and absence of SCRIB KD.

- In response to this concern, we have included the data of IC50 values in parent, tamoxifen resistant and SCRIB-KD tamoxifen resistant cells as commented in #12. The data clearly showed that SCRIB and SLC3A2 regulate tamoxifen resistant phenotypes in ER+ breast cancer cells.

Reviewer #2 (Remarks to the Author):

The manuscript submitted by Saito et al. describes a role for SCRIB in regulating amino acid

metabolism in ER+ breast cancer cells. The study is a follow-up to the group's recent finding that LLGL2 impacts cell proliferation under nutrient stress and regulates tamoxifen resistance. Overall, this is a novel study that adds to our understanding of mechanisms underlying metabolic functions in breast cancer growth, with a strong link to the cell polarity machinery. The data presented are generally convincing and clear.

The authors demonstrate that knockdown of SCRIB affects global metabolic profiles and associated with SLC3A2 via the amino-terminus and forms a quaternary complex with LLGL2 and SLC7A5. Is SCRIB/SLC3A2 associated specifically with LLGL2 and SLC7A5, or are there other transporter complexes that SCRIB associates with? For example, SLC3A2 also associates with SLC7A11 (PMID: 16103098). The dramatic changes in metabolism with SCRIB-KD (i.e. almost all metabolic changes are associated with metabolite down-regulation in SCRIB-KD cells) suggest that this may be the case.

- This is an interesting and important question. We have examined whether SCRIB could interact with SLC7A11 via SLC3A2. We overexpressed HA-SCRIB, SLC3A2 and Flag-SLC7A11 in HEK293T cells and performed immunoprecipitation using anti-HA antibody. The immunoprecipitants were analyzed with immunoblot using anti-HA, anti-SLC3A2, and anti-Flag M2 antibodies. Interestingly, we observed the complex formation among SCRIB, SLC3A2, and SLC7A11, suggesting that SCRIB could associate with SLC7A11 in cells (Fig. 3e). However, the expression levels of SLC7A11 are quite low in ER+ breast cancer cells than in basal breast cancer cells (Fig. 3f). Thus, we think that SCRIB mainly regulates the function of SLC7A5 in ER+ breast cancer and the functional contribution of SLC7A11 in ER+ breast cancer cells could be less to the metabolic change in SCRIB-KD cells although further investigation will absolutely be needed. Furthermore, SLC3A2 has been reported to interact with CD147 (PMID: 15901826) and integrin (PMID: 11121428). Further investigation will be needed to confirm the contribution of these molecules in the SCRIB-dependent cell metabolisms in ER+ breast cancer cells.

The finding that SCRIB expression is regulated by MYC and implicated in tamoxifen resistance is of interest. While SCRIB is upregulated in TamR cells and SCRIB-KD and SLC3A2-KD cells have impaired growth, the conclusion that SCRIB-SLC3A2 are regulators of tamoxifen resistance is an over-interpretation of this data. SCRIB-KD has the same effect in TamR as in normal MCF7 cells. This could be addressed by determining if SCRIB and SLC3A2 knockdown re-sensitize cells to tamoxifen.

- We thank constructive comments. In response to this concern, we investigated the IC50 values of tamoxifen in parental, tamoxifen-resistant, SCRIB-KD tamoxifen-resistant, and SLC3A2-KD tamoxifen-resistant cells. As shown in Figure 5, both SCRIB-KD and

SLC3A2-KD sensitized resistant cells to tamoxifen. These results strongly support that SCRIB-SLC3A2 signaling axis contributes to the regulation of tamoxifen resistant phenotypes in MCF-7 cells.

I am enthusiastic about this manuscript, however these and additional points need to be addressed. Specific points:

1. Figures 1a, 1e, 4a, 4b, – what does Probability on the y-axis represent? Overall survival, recurrence-free survival? Other?

- In response, the title on y-axis “probability” indicates “relapse-free survival”.

2. Figures 1c, 4d, 4e - What does each dot represent? Are these independent experiments, or different wells from the same experiment? This information should be provided in the figure legends.

- In response to this issue, we had included the sentences in the figure legends. Each dots represent the relative luminescence intensity from different wells from the same experiment. We performed three independent experiments with 9 technical replicates and one representative data is shown in Figure 1c and e.

3. All of the graphs for sphere growth (Fig. 1c, e) use relative ATP as the measurement. My concern is that this is a metabolic readout for growth. ATP-levels are an indirect measurement of growth and are affected by SCRIB-KD (Supp. Fig 2). Therefore, can an independent measurement of sphere growth should be used to validate the results obtained with the ATP assay? At least for some of the data.

- In response to this concern, we performed cell proliferation assay by counting alive and dead cell numbers with trypan blue staining. As shown in Supplementary Fig. 1k, 1l, 1m, and 1n, the results of proliferation assay by cell counting are consistent with the data of celltiter glo 3D assay measuring ATP amount. Thus, we concluded that relative ATP amount is equivalent to the growing cell number in our experiments.

- In addition, to avoid the misleading of growth data, the title of y-axis shown as “relative ATP” is changed to “relative luminescence”.

4. Some of the SLC3A2 blots have a large number of bands or smears and are difficult to interpret (e.g. Fig. 2g, TCL; 2h TCL; 5h).

- In response to this concern, we have included the figure that explains the multiple bands and smears of SLC3A2 in Supplementary Fig. 3d. As previously reported in Fort et al. (J. Biochem. 282; 31444-31452, 2007; doi:10.1074/jbc.m704524200), immunoblot image of SLC3A2 shows multiple bands due to the modification of glycosylation. Monomer (glycosylated form and non-glycosylated form) and proteolytic fragment are indicated in Supplemented Fig. 3d.

5. Fig. 5f – the blot should include a blot for myc. I expect it would be upregulated in addition to

SCRIB. Also, does MYC knockdown in TamR MCF7 cells block upregulation of SCRIB protein?

- In response to this concern, we have included the blot image for MYC in Fig. 5g. We found that MYC expression is upregulated in TamR MCF-7 cells compared with the parental MCF-7 cells. Also, we performed MYC knockdown in TamR MCF-7 cells and found that MYC-KD reduced the protein levels of SCRIB in TamR MCF-7 cells (Fig. 5h).

6. How was the metabolomics data normalized? I did not see this information in the figure legend or methods section. Also, what are the criteria/cut-offs used for determining down-regulation?

- In response to this concern, we have included the sentences on data normalization in the method section. All metabolite concentrations were normalized with cell numbers for cultured cells or with the weight for mouse tumor tissues.

- All metabolites' data were statistically analyzed by two-tailed student's t-test using MetaboAnalyst 5.0 (<https://www.metaboanalyst.ca>). The metabolites which showed statistical significance by t-test are described as down-regulated or up-regulated metabolites in this manuscript.

REVIEWERS' COMMENTS:

Reviewer #1 (Remarks to the Author):

The authors have thoroughly addressed all of my points.

Reviewer #2 (Remarks to the Author):

The authors have provided substantial new data and have addressed all of my previous concerns. This paper represents an important step towards understanding tamoxifen resistance in ER+ breast cancer cells.